# Peer review of "Unveiling Cryptocurrency Impact on Financial Markets and Traditional Banking Systems: Lessons for Sustainable Blockchain and Interdisciplinary Collaborations"

_jrfm, doi:10.3390/jrfm17020058_

Round 1

Reviewer 1 Report

Comments and Suggestions for Authors

Introduction

- Clearly states aims to analyze impact of cryptocurrencies and blockchain on financial systems in UK and USA

- Objectives need further refinement and specificity

- Background should define key concepts of cryptocurrencies and blockchain to orient readers

Literature Review

- Effectively reviews prior research on opportunities and challenges posed by cryptocurrencies and blockchain innovations

- Needs better organization of sources under coherent themes

- Tends to present existing research uncritically rather than assessing limitations, disagreements or gaps

- Needs to clearly map connections to present work and spell out unique contributions

Methodology

- Qualitative secondary research design clearly described including data sources and thematic coding used

- Justification for geographical focus on UK and USA requires strengthening 

- Details needed on sample sizes supporting statistical analysis

- Discussion of ethical considerations limited to citations - social and policy implications need addressing

Results & Analysis

- Major highlight is comparative analysis of adoption metrics, integration techniques and regulatory approaches in UK vs USA

- Testing statistical significance would substantiate claims of differences

- More examination needed of political vs just innovation-based factors behind policy variations

Conclusions & Implications

- Conclusion summarizes key findings and macro-level implications regarding changing financial environments 

- Practical implications for specific stakeholder groups need elaboration

- Future research directions lack detail on critical gaps and new priority questions

In summary, I have recognized the valuable contributions of the manuscript in reviewing literature and comparing UK and USA developments. Addressing the comments through revisions would enhance quality before acceptance.

Reviewer 2 Report

Comments and Suggestions for Authors

Unveiling Cryptocurrencies Impact on Financial Markets and Traditional Banking System: Lessons for Sustainable Blockchain and Interdisciplinary Collaborations addresses the role of cryptocurrencies in the US and UK is something between a survey article and literature review.  The paper covers a range of important topics, but the wide scope of the article causes the authors to overly simplify concepts based on the author’s perspectives. 

For example, the text states “New businesses are increasingly 75 leaning towards the concept of "Initial Coin Offering" (ICO) as opposed to the more conventional "Initial Public Offering" (IPO) (Masiak et al., 2019), despite the fact that traditional corporate governance standards may still have some influence.”  Aside from tokenised securities, the role of crypto and stocks are quite different. This is far too much of a simplification that either requires greater context or should be removed. This also connects to the issue of defining a cryptocurrency. 

There are a range of virtual assets, and the term cryptocurrency can be used to discuss the full range, including assets (like governance or utility tokens) that are not intended to be used as currencies or it can refer specifically to assets that function as currencies.  Even stable coins, which are intended to function as currencies, are usually centralised and therefore argued by some not to be cryptocurrencies.  It’s important to clearly specify the types of crypto assets being discussed.  By discussing the UK and US, the authors are viewing jurisdictions where these assets are primary used for investment and not usage.  The authors note ““Bitcoin, are a new kind of investment instrument that defies established conventions…”. However, this is not the case in all countries.  As such, discussions about global regulation are viewed through a limited lens.

Suggestions:

A more balanced discussion that includes necessary context to the issues discussed and counterarguments would add depth to the article. Also, it would be beneficial to cite a broader range of resources to support claims and provide a more comprehensive view of the topic.

Page 3 has a very long paragraph that could be split for readability. 

I enjoyed the article and hope these suggestions help.  

Author Response

Please see teh attachedment.
